# Improvement of Learning and Memory in Senescence-Accelerated Mice by S-Allylcysteine in Mature Garlic Extract

**DOI:** 10.3390/nu12061834

**Published:** 2020-06-19

**Authors:** Masakazu Hashimoto, Tsuyoshi Nakai, Teruaki Masutani, Keiko Unno, Yukihiro Akao

**Affiliations:** 1United Graduate School of Drug Discovery and Medical Information Sciences, Gifu University, 1-1 Yanagido, Gifu 501-1193, Japan; x3501003@edu.gifu-u.ac.jp; 2Research and Development Department, Ichimaru Pharcos Co., Ltd., 318-1 Asagi, Gifu 501-0475, Japan; t.nakai1126@gmail.com (T.N.); masutani-teruaki@ichimaru.co.jp (T.M.); 3School of Pharmaceutical Sciences, University of Shizuoka, 52-1 Yada, Shizuoka 422-8526, Japan; unno@u-shizuoka-ken.ac.jp

**Keywords:** S-allylcysteine (SAC), mature garlic extract (MGE), hippocampal neuron, senescence-accelerated mice, memory, cognitive function

## Abstract

S-allylcysteine (SAC), a major thioallyl compound contained in mature garlic extract (MGE), is known to be a neuroactive compound. This study was designed to investigate the effects of SAC on primary cultured hippocampal neurons and cognitively impaired senescence-accelerated mice prone 10 (SAMP10). Treatment of these neurons with MGE or SAC significantly increased the total neurite length and number of dendrites. SAMP10 mice fed MGE or SAC showed a significant improvement in memory dysfunction in pharmacological behavioral analyses. The decrease of α-amino-3-hydroxy-5-methyl-4-isoxazolepropionic acid (AMPA) receptor, *N*-methyl-d-aspartate (NMDA) receptor, and phosphorylated α-calcium/calmodulin-dependent protein kinase II (CaMKII) in the hippocampal tissue of SAMP10 mice fed MGE or SAC was significantly suppressed, especially in the MGE-fed group. These findings suggest that SAC positively contributes to learning and memory formation, having a beneficial effect on brain function. In addition, multiple components (aside from SAC) contained in MGE could be useful for improving cognitive function by acting as neurotrophic factors.

## 1. Introduction

In recent years, the consumption of certain foods by aged individuals for the purpose of promoting and maintaining brain function has attracted much attention, particularly in terms of improving the accuracy of memory and judgment. Above all, garlic (*Allium sativum* L. *Liliaceae*) has been widely used as a food and medicine for thousands of years [1,2]. Garlic contains S-allylcysteine (SAC), which is considered to be useful for memory improvement. Mature garlic extract (MGE) made from garlic that has been aged at a low temperature contains more SAC than aged-garlic extract (AGE) made from common black garlic. In addition, MGE contains cycloalliin, which is useful for increasing fibrinolytic activity and preventing hyperlipidemia [3,4], and γ-glutamyl-S-allylcysteine, which contributes to hypotensive effects through angiotensin converting enzyme inhibitory and vasodilating activities [5,6].

SAC, which is an organosulfur compound in garlic, has a high bioavailability of 98.2% (rat, 50 mg/kg, p.o.) [7]. Therefore, by reaching the systemic circulation and passing through the blood–brain barrier [8], SAC can have various effects on the brain. SAC has been confirmed to cause a significant increase in the formation of branching per axon as well as a survival-promoting effect on primary cultured hippocampal neurons [9,10]. Since such neurotrophic factors improve learning and reduce memory impairment [11], SAC is considered to be a beneficial component for brain function. Also, SAC has been shown to have a selectively neuroprotective effect by reducing cell death caused by endoplasmic reticulum stress induced by amyloid β (Aβ) and tunicamycin [12,13,14]. In addition, it has been found that SAC inhibits Aβ fibrillation, destabilizes preformed Aβ fibrils [15], and reduces hyperphosphorylation of the tau protein, which induces neurofibrillary tangles and Aβ deposition [16]. Therefore, SAC is expected to be applied for the treatment of neurodegenerative diseases such as Alzheimer’s and Parkinson’s diseases.

Senescence-accelerated mice (SAM) have been established as a model for studying human aging and age-related disorders. There are several senescence-prone inbred strains (SAMP) and senescence-resistant inbred strains (SAMR). SAMP mice have a short lifespan and exhibit many characteristic age-dependent pathologies at an early age [17,18]. Among these strains, the SAMP10 mouse strain was established by Shimada and colleagues [19,20]. The age-related morphological changes seen in the SAMP10 brain, such as the retraction of dendritic arbors, a decrease in the density of the dendritic spine [21], a loss of synapses [22], and impairment in learning and memory [23,24,25,26], are more consistent with observations on the aging human brain than those on the brain of mice with Alzheimer’s disease. Several behavioral tests of brain function using SAMP10 and SAMR1 mice have been widely used to study the effects of food materials on the prevention of brain senescence, and materials such as garlic [27,28] and green tea [29,30,31] have been found to improve learning memory impairment and suppress brain atrophy.

Regarding the physiological function of SAC and AGE components, the research focus has been on the antioxidative effects while the neurotrophic factor is poorly understood. Although there have been many reports that AGE and SAC have useful effects on neuronal morphological changes and learning behavior [9,10,27,28], the detailed mechanism of how SAC affects memory-related receptors, such as the α-amino-3-hydroxy-5-methyl-4-isoxazolepropionic acid (AMPA) receptor and the *N*-methyl-d-aspartate (NMDA) receptor, in the hippocampus is not clear. In addition, few reports have examined the effects of long-term intake of these substances on learning and memory. In this study, we examined the efficacy of MGE, but not AGE, and SAC for increasing the total neurite length and the number of dendrites in primary cultured embryonic mouse hippocampal neurons. Next, we analyzed learning and memory-formation-related behavioral experiments and protein expression levels in the hippocampal tissue of SAMP10 mice that were continuously fed a diet containing MGE or SAC for a period of 10 months (from ages 2 to 11 months). Our findings demonstrate strong evidence that MGE and SAC possess potential neurotrophic properties and also preserve learning and memory functions to maintain young brain function.

## 2. Materials and Methods

### 2.1. Animals and Preparation of Matured Garlic Extract

C57BL/6J mice were obtained from Charles River Laboratory Japan (Yokohama, Japan).

Male SAMP10/TaSlc (SAMP10) and male SAMR1/TaSlc (SAMR1) mice were obtained from Japan SLC (Shizuoka, Japan) at 4 weeks of age. Mice were housed under a standard 12 h light/dark cycle (light phase 9:00–21:00) at a constant temperature of 22 ± 1 °C with food and water provided ad libitum throughout the experiments.

SAMP10 and SAMR1 mice were fed a diet (CE-2; Clea, Tokyo, Japan) containing MGE or SAC (Tokyo Kasei, Tokyo, Japan) starting at 2 months of age. MGE was manufactured by extracting the water-soluble fraction of garlic supplied by Takko Kawamura Agri Service Inc. (Aomori, Japan).

In this study, a diet with a low concentration of MGE, i.e., 0.20% of the diet (*w*/*w*) (L-MGE) and one with a high concentration of MGE, i.e., 1.0% of the diet (*w*/*w*) (H-MGE) were prepared. In addition, a diet containing the same amount of SAC as in H-MGE was prepared. In a previous report, a diet containing 2% AGE or 0.002–0.004% SAC was used [27,32]. Based on that, we set the amount of SAC to 0.002% (=20 mg/kg diet) and the amount of H-MGE to 1.0%. L-MGE was set to 0.2% to investigate the concentration-dependent effect. Since mice consume 150 g of diet/kg (body weight)/day [33], L- and H-MGE were consumed at 0.30 g and 1.5 g MGE/kg (body weight)/day, respectively. As a result of quantifying the amount of SAC by high performance liquid chromatography (HPLC) (Shimadzu, Tokyo, Japan), 1.0 g of MGE contained 2.0 mg of SAC. Therefore, the L-MGE- and H-MGE-fed groups consumed 0.60 and 3.0 mg of SAC per day, respectively.

### 2.2. Cell Culture

Primary cultured hippocampal neurons were prepared from C57BL/6J mice on gestational days 15–16, as described previously with minor modifications [34]. The embryonic hippocampus was digested with 0.25% trypsin and 0.1 mg/mL DNase for 10 min at 37 °C and then gently pipetted to mechanically dissociate the cells. Neurons were seeded on poly-d-lysine-coated cell disks (Sumitomo Bakelite, Tokyo, Japan) in neural basal medium containing B-27 and GlutaMax supplement (Thermo Fisher Scientific, Waltham, MA, USA), and the cultures were started on day 0 in vitro (DIV 0). Culture medium was exchanged for fresh medium every 3–4 days. MGE or SAC was added to the culture medium along with 1 μM cytosine β-d-arabinofuranoside on DIV 2. At DIV 4 (48 h) and 5 (72 h), the total neurite length and number of dendrites treated with MGE or SAC were measured by immunofluorescence staining.

### 2.3. Immunofluorescence and Image Quantification

Hippocampal neurons on poly-d-lysine-coated cell disks were fixed with formaldehyde for 10 min and then incubated with blocking buffer (PBS with 10% goat serum and 1% BSA) for 1 h at room temperature on DIV 4 and 5. Anti-MAP2 antibody (dendrite marker, Abcam, Cambridge, UK) was added to the disks at 1:1000 dilution in Can Get Signal Solution B (Toyobo, Osaka, Japan), which were then incubated for 1 h at room temperature. After 2 washings with TBS-T solution, each for 10 min, goat-rabbit IgG antibody coupled to Alexa fluor 568 (Thermo Fisher Scientific, Waltham, MA, USA) at 1:200 dilution in Can Get Signal Solution B was added, and the cells were incubated for 30 min at room temperature under shaded conditions. After 3 washings with TBS-T solution for 10 min each time, nuclear DNA was stained with Hoechst 33342 (Dojindo, Kumamoto, Japan) at 1:1000 dilution in distilled water. After another 3 washings for 10 min each time, the cell disks were mounted on glass slides. Images were acquired with a fluorescence microscope (Olympus, Tokyo, Japan). Images of MAP2-positive cells obtained by immunofluorescence staining were transformed using an IN Cell Translator (GE Healthcare, Buckinghamshire, UK), and quantification of total neurite length and number of dendrites per neuron was performed with an IN Cell Analyzer Workstation (GE Healthcare, Buckinghamshire, UK).

### 2.4. Behavioral Experiments

SAMP10 mice were randomly divided into four groups (*n* = 18 mice per group). The mice were fed a CE-2 diet containing MGE or SAC for a period of 10 months (from ages 2 to 11 months). One of the SAMP10 groups and the SAMR1 group (*n* = 12) were fed the CE-2 diet without MGE or SAC and acted as control mice for the behavioral experiments. Six months after the start of breeding, additional 4-week-old SAMP10 mice were purchased as Young-SAMP10 mice to allow us to observe age-related declines in brain function. Learning and memory ability were measured by performing the Y-maze, step-through passive avoidance, and novel object recognition tests on animals at 11–12 (Old-SAMP10 and Old-SAMR1) or 5–6 (Young-SAMP10) months of age. The mice were sacrificed at the age of 12 (Old-SAMP10 and Old-SAMR1) or 6 (Young-SAMP10) months, and tests were carried out to obtain the hippocampal tissue. The samples were immediately frozen at −80 °C. All protocols for animal procedures were approved by the University of Shizuoka Laboratory Animal Care Advisory Committee (approval No. 166197) in accordance with the Internal Regulations on Animal Experiments at the University of Shizuoka, which are based on the Law for the Humane Treatment and Management of Animals (Law No. 105, 1 October 1973).

#### 2.4.1. Working Memory

Spontaneous alternations and exploratory behavior in the Y-maze were used as hippocampal-based tasks to assess working memory. Immediate working memory was evaluated by recording spontaneous alternations during a single session in the Y-maze [35,36]. The Y-maze apparatus was made of black plastic with three arms (40 cm × 15 cm × 35 cm), each extending at 120° from a central platform. Each mouse was placed on the end of one arm and allowed to move freely through the maze during an 8-min session, and the number of arm entries was counted. Each series of arm entries was visually recorded, and an arm entry was defined as when the hind paws of the mouse were completely within the arm. The number of alternations was defined as the number of combinations (i.e., abc, bca, triplets) of entrances into the three different arms in succession and was considered to reflect the working memory capacity. The percentage of spontaneous alternations (%) was calculated from the following formula and used as an index of short-term memory:Alternation (%) = (number of alternations)/(total arm entries − 2) × 100.

#### 2.4.2. Memory Acquisition Test and Retention Test

A step-through passive avoidance task was carried out according to the protocol method reported earlier [29]. This test was based on the fact that mice prefer dark places. The apparatus was connected to a light chamber and a dark chamber with a door between them. The mice in the test were initially placed in the light chamber. When a mouse entered the dark chamber, the door was closed, and an electric foot-shock was delivered at 0.05 mA for 1 s (Muromachi Kikai Co. Ltd., Tokyo, Japan). The mouse was then gently removed and replaced in the bright room. One minute later, the door was opened, and the time taken for the mouse to enter the dark chamber was measured. The trial was terminated when the mouse remained in the light chamber for 300 s without entering the dark room, and this was repeated five times until the mouse had satisfied the acquisition criterion. In such multiple-trial passive avoidance tests, the number of trials required for the mouse to satisfy the acquisition criterion is usually regarded as an index of memory acquisition. The total time spent in the light chamber during each trial was deducted from 300 s and was considered the time needed for learning. The time taken for each trial was totaled—the shorter the learning time, the higher the learning ability.

One month later, the mice were assessed again to see whether they remained in the light chamber. The number of mice remaining in the light chamber for 300 s was used as the acquisition criterion for long-term memory.

#### 2.4.3. Novel Object Recognition Test

This task was performed on days 1–5 according to a previously described protocol with some modifications [37,38]. The novel object recognition test was based on the characteristic of a preference for a novel object. The task was divided into three different sessions (habituation, training, and retention). In the habituation session, each mouse was individually placed in an open box (30 cm × 30 cm × 35 cm height) without objects for three consecutive days and allowed to explore for 10 min each day. Secondly, a training session was performed on the day 4. Two novel objects (X and Y) were placed in the open box, and the mice were allowed to explore the objects freely for 10 min. The total time spent exploring an object, which was calculated as the total time that a mouse directed their nose toward an object at a distance of <1 cm and/or touched the object with their nose, was assessed manually for 10 min using two stopwatches. Thirdly, a retention session was performed on day 5. The mice were allowed to explore an open field for 10 min in the presence of two objects of different shapes and colors, i.e., the familiar object X and a novel object, Z.

The time spent exploring each object was recorded as before. Then, the exploration time for each object in the training (X or Y) and retention (X or Z) sessions was evaluated against the total exploring time. Cognitive function was evaluated by exploratory preferences obtained from the time ratio for each object, e.g., X or Y/(X + Y) × 100 (%) in the training session, and X or Z/(X + Z) × 100 (%) in the retention session. An exploratory preference index of 50% corresponds to chance, and a significantly higher exploratory preference index reflects good recognition memory.

#### 2.4.4. Western Immunoblotting

At the end of the behavioral experiments, the hippocampus was removed, placed on an ice-cold plate, immediately frozen, and stored −80 °C. Hippocampus protein extracts were obtained by homogenization of the hippocampal tissue in Passive Lysis Buffer (Promega, Madison, WI, USA) supplemented with proteinase and phosphatase inhibitors. The homogenate was centrifuged at 13,000× *g* for 20 min to obtain a supernatant, which was then subjected to protein estimation (Bradford assay), and a defined volume of the supernatant containing a fixed amount of protein was analyzed by Western immunoblotting.

For preparation of the tissue lysates for Western blotting, a Laemmli buffer (Bio-Rad Laboratories, Hercules, CA, USA) was added. Prior to electrophoresis, samples were denatured at 95 °C for 6 min. An equal amount of total protein (20 μg) from tissue homogenate was loaded onto a 7.5% mini-gel (Mini-PROTEAN TGX Precast Gel, Bio-Rad Laboratories, Hercules, CA, USA) along with a molecular weight marker (Bio-Rad Laboratories, Hercules, CA, USA). Protein bands on the separating gel were transferred to a polyvinylidene difluoride membrane (Trans-Blot Turbo Mini PVDF, 0.2 μm, Bio-Rad Laboratories, Hercules, CA, USA) in accordance with the manufacturer’s instructions. The membranes were then blocked for 1 h in blocking buffer (PVDF Blocking Reagent for Can Get Signal, Toyobo, Osaka, Japan) at room temperature and incubated in solution 1 (Can Get Signal Solution, Immunoreaction Enhancer Solution for primary antibody, Toyobo, Osaka, Japan) and the primary antibodies AMPA receptor GluR1 subunit (anti-GluR1; molecular weight (MW), 102 kDa; 1:1000 dilution; Abcam, Cambridge, UK), anti-GluR1 phosphorylated at serine 831 (anti-pGluR1; MW, 106 kDa; 1:1000 dilution; Abcam, Cambridge, UK), NMDA receptor 2B subunit (anti-NR2B; MW, 166 kDa; 1:1000 dilution; Abcam, Cambridge, UK), anti-NR2B phosphorylated at tyrosine 1472 (anti-pNR2B; MW, 180 kDa; 1:1000 dilution; Merck, Darmstadt, Germany), and anti-α-calcium/calmodulin-dependent protein kinase II phosphorylated at threonine 286 (anti-pCaMKII; MW, 50 kDa; 1:1000 dilution; Cell Signaling Technology, Danvers, MA, USA) overnight at 4 °C. Membranes were washed with Tris-buffered saline with 0.1% Tween-20 (TBS-T) and then incubated in solution 2 (Can Get Signal Solution, Immunoreaction Enhancer Solution for secondary antibody, Toyobo, Osaka, Japan) and the secondary antibody (HRP-Linked Anti-IgG, 1:10,000 dilution, GE Healthcare, Buckinghamshire, UK) for 1 h at room temperature. After the membrane had been washed with TBS-T, the relative amounts of bound antibodies were detected with a chemiluminescent substrate (ECL, GE Healthcare, Buckinghamshire, UK). The specific bands were scanned and quantified with ChemiDoc XRS+ and ImageLab software (Bio-Rad Laboratories, Hercules, CA, USA).

### 2.5. Flowchart

The summarized method is provided in the following flowchart (Figure 1).

### 2.6. Statistical Analyses

The results were analyzed using JMP 8 (SAS Institute Inc., Cary, NC, USA). The data were collected from at least three independent experiments and are expressed as the mean ± standard error of the mean (SEM). For all results, assuming a Gaussian distribution, data were analyzed by one-way analysis of variance (ANOVA). To perform multiple comparisons, Dunnett’s test or the Tukey–Kramer test were used for post-hoc analysis after ANOVA. *p* < 0.05 was considered statistically significant. The levels of statistical significance are indicated as follows: *; *p* < 0.05, **; *p* < 0.01.

## 3. Results

### 3.1. Effects of MGE on the Total Neurite Length and Number of Dendrites in Primary Cultured Hippocampal Neurons

We first examined whether MGE and SAC induce increases in the total neurite length and number of dendrites in hippocampal neurons. The total neurite length and number of dendrites per neuron at 0 h (DIV2) were 212.5 ± 6.08 μm (Table 1) and 3.47 ± 0.127 (Table 2), respectively. A typical image of a MAP2-positive neuron treated with 50 μg of MGE is shown in Figure 2. Primary cultured hippocampal neurons treated with MGE at 48 and 72 h showed significant increases in the total neurite length and number of dendrites, and the neurons increased in a concentration-dependent manner at 48 h. In the case of treating with SAC, the neurite length and number of dendrites increased significantly; however, no concentration dependency was found. The concentration of SAC that showed the maximum effect was 10 ng/mL at 72 h. These results suggest that SAC and the multiple other components contained in MGE have the ability to synergistically enhance the effect of early neurite outgrowth.

### 3.2. Y-Maze Test

The Y-maze test was performed to investigate the effects of MGE and SAC on the improvement of the spatial working memory, which is a kind of short-term memory. As shown in Figure 3A, old SAMP10 mice showed significantly reduced spontaneous alternations compared with old SAMR1 and young SAMP10 mice (*F* (2, 33) = 17.33, *p* = 7.17 × 10^−6^). In old SAMP10 mice, the number of alternations in the MGE- and SAC-fed groups significantly increased compared with that of the control group (*F* (3, 57) = 23.99, *p* = 3.62 × 10^−10^; Figure 3B). As shown in Figure 3C, young SAMP10 mice showed a high value for total arm entries (*F* (2, 33) = 40.35, *p* = 1.37 × 10^−9^), although there was no significant difference in the total arm entries in old SAMR1 and old SAMP10 mice. On the other hand, there was no significant difference in the total arm entries among the SAMP10 groups (*F* (3, 57) = 5.868, *p* = 1.46 × 10^−3^; Figure 3D). These results suggest that the neural circuit for memory formation in old SAMP10 mice is stimulated by MGE and SAC intake and has a positive effect on recovery from the decline in short-term memory ability. In addition, only young SAMP10 mice showed a high rate of spontaneous locomotor activity, which was calculated from the number of arm entries.

### 3.3. Step-Through Passive Avoidance Test

The time taken to learn not to enter the dark chamber was recorded using a step-through passive avoidance task, in which a shorter learning time implied a higher learning ability. As shown in Figure 4A, the learning time of old SAMP10 mice fed a normal diet was significantly longer than that of young SAMP10 and old SAMR1 mice (*F* (2, 33) = 18.38, *p* = 4.32 × 10^−6^). In the old SAMP10 mice fed MGE or SAC, the learning times were significantly shortened (*F* (3, 56) = 25.99, *p* = 1.15 × 10^−10^; Figure 4B), equivalent to the learning ability of young SAMP10 mice. These results suggest that MGE and SAC contribute to learning acquirement and efficiency in old SAMP10 mice.

The memory retention test showing the avoidance response was assessed at 1 month after the acquisition test. The number of mice that stayed in the light chamber for at least 300 s was only measured once. As shown in Table 3, all of the young SAMP10 mice succeeded at remembering the avoidance response (8/8), whereas only 46.7% (7/15) of the old SAMP10 mice remained in the light chamber. An increased number of old SAMP10 mice in the MGE- or SAC-fed groups satisfied the acquisition criteria, exceeding the success rate of the old SAMR1 mice (Table 4). These findings suggest that long-term spatial memory retention is possibly due to intake of MGE and SAC.

### 3.4. Novel Object Recognition Test

The non-spatial memory ability was also evaluated in the novel object recognition test. During the training session, there was no biased exploratory preference in all groups (Figure 5A,B). When the retention session was performed 24 h after the training session, old SAMP10 mice fed a normal diet did not change their level of exploratory preference, whereas young SAMP10 and SAMR1 mice showed significantly increased preference for novel object C (*F*(2, 31) = 9.299, *p* = 6.86 × 10^−4^; Figure 5C). SAMP10 mice in the MGE- and SAC-fed groups also showed a significantly increased exploratory preference (*F* (3, 52) = 6.788, *p* = 6.00 × 10^−4^; Figure 5D), indicating that a diet containing MGE and SAC supports non-spatial memory formation in SAMP10 mice. These results suggest that the diets containing these ingredients reduced age-related memory decline and enhanced long-term non-spatial memory retention. In addition, the changes in exploratory preference did not show a concentration-dependent characteristic in the MGE-fed group.

### 3.5. Western Immunoblotting

Several studies have indicated that tyrosine phosphorylation of NR2B and serine phosphorylation of GluR1 can regulate the activity of NMDA and AMPA receptors in neurons and that these phosphorylation reactions are involved in learning and memory formation [39,40]. It is also well known that CaMKII-dependent signaling in neurons is involved in survival, brain development, learning, and memory formation [40,41,42,43,44]. Therefore, the relationship between protein expression and learning and memory formation was investigated using hippocampal tissue for Western immunoblotting after completion of the behavioral experiments (Figure 6A). The protein expression levels of learning and memory-related receptors and the phosphorylation levels (GluR1, pGluR1, NR2B, pNR2B, and pCaMKII) were reduced in old SAMP10 mice compared with the expression and phosphorylation levels in young SAMP10 mice (Figure 6B). Old SAMP10 mice fed H-MGE showed significant suppression of the decrease in the expression levels of all proteins (Figure 6C). On the other hand, the SAC-fed group only showed suppression of the decrease in the expression of GluR1 and pCaMKII, although no concentration-dependent actions of MGE and no correlation between the H-MGE and SAC groups were observed. These results suggest that SAC and the multiple other components contained in MGE have the ability to increase protein expression, suggesting that they work to maintain and enhance learning and memory functions.

## 4. Discussion

The essence of the brain is the processing of external information and subsequent plastic regulation of neuronal function. Neurons form networks through synaptic structures and communicate with each other through neurotransmitters and synaptic receptors [45]. In this study, treatment of primary cultured hippocampal neurons with MGE or SAC significantly increased the total neurite length and number of dendrites. SAC showed a certain neurotrophic effect and further activated neurons through a synergistic effect with multiple other components contained in MGE. We suggest that the neurotrophic effects of MGE and SAC are strongly involved in the enhancement of transmission efficiency and information-processing ability by neural network formation. Furthermore, we propose that the increase in the number of dendrites plays a role in the ability of the brain to receive a lot of information. Thus, although SAC showed the maximum effect on the neurons at low concentrations, these data (Table 1 and Table 2) alone do not suggest that it has a useful effect on learning and memory.

Therefore, pharmacological behavioral tests were performed to investigate the inhibitory effects on memory dysfunction of feeding a diet containing MGE or SAC to SAMP10 and SAMR1 mice. The Y-maze is considered to be a hippocampus-dependent memory test, since it evaluates spatial working memory, an index of short-term memory, through assessing the continuous selection of arms [46,47]. In addition, it has the advantage of providing a measure of locomotor activity of mice by counting the number of arm entries [48]. The step-through passive avoidance test is closely related to the amygdala-dependent memory, since it is used as an index of long-term memory to evaluate avoidance behavior against an aversive stimulus (electric foot-shock) that has been experienced once [49]. In addition, the novel object recognition test evaluates long-term memory through the recognition of unique non-spatial information of novel objects by utilizing a characteristic of rodents, i.e., the preference for novelty. In a previous study, this test was related to olfactory-cortex-dependent memory, which is considered to be one of the major pathways of neural information related to episodic memory [50]. It has been suggested that the hippocampus reactivates specific memory representations of the olfactory cortex and amygdala during memory retrieval [51], and, in particular, the amygdala and hippocampus collect information from related cortical areas and are deeply involved in the processes of cognition, memory formation, and emotional expression [52]. In all pharmacological behavioral tests performed, SAMP10 mice fed MGE or SAC showed significantly reduced learning and memory dysfunction and significant improvements in learning and short- and long-term memory formation. In addition, since the locomotor activity obtained from the number of arm entries did not differ significantly in any group, there were no differences in the amount of exercise, exploratory behavior, or motivation. Short-term memory formation requires the activity of existing ionotropic receptors and kinases in neuronal cells, and memory consolidation from short- to long-term memory requires the induction of de novo protein synthesis in the brain after learning [53,54]. Therefore, it is suggested that, by improving the above-mentioned process, MGE and SAC enhance learning activity and memory consolidation.

It is known that synaptic transmission efficiency is not constant and changes following exposure to a stimulus, a phenomenon called synaptic plasticity [55]. The long-term memory circuit is formed by the induction of long-term potentiation (LTP), which causes a long-term increase in transmission efficiency at neuronal synapses and requires activation of NMDA-type receptors and induction of AMPA-type receptor expression [38,39,56,57]. The AMPA-type receptor has four subunits, GluR1–4 [58], and the regulation of AMPA-type receptor expression is one of the important mechanisms underlying synaptic plasticity. Similarly, the NMDA-type receptor is composed of NR1 and NR2 subunits, and the NR2 subunit has four subtypes, NR2A–2D. LTP is induced by the activation of CaMKII caused by Ca^2+^ influx from NMDA receptors. Activated CaMKII phosphorylates GluR1 at serine 831, increases the channel conductance states, and is involved in LTP induction [43,59]. LTP is thought to be induced by an increase in an excitatory postsynaptic current (EPSC) and synaptic plasticity when phosphorylated GluR1 is recruited onto the postsynaptic membrane to increase the synaptic transmission efficiency [38,39,56,60]. Thus, the AMPA-type receptor containing GluR1 must be expressed on synaptic membranes for memory formation, and the activation of the NMDA-type receptor that stimulates them is essential for this process. However, there have been numerous reports of decreases in AMPA- and NMDA-type receptor expressions in the hippocampus of SAMP10 and aged rodents, suggesting an association with an age-related decline in learning ability [61,62,63,64,65,66]. It has also been reported that one of the earliest biological manifestations of Alzheimer’s dementia is a decrease in AMPA-type receptors and impaired synaptic plasticity [67,68], and a reduction of autophosphorylation of CaMKII at threonine 286 in the frontal cortex and hippocampus of Alzheimer’s disease brains is a key contributor to synaptic dysfunction, neurodegeneration, and memory impairment [44,69]. We analyzed the hippocampal proteins of SAMP10 after the behavioral experiments and found that the expression levels of GluR1, NR2B, and phosphorylated CaMKII, which are involved in learning and memory abilities, were significantly increased in the MGE-fed group of SAMP10 mice. On the other hand, the SAC-fed group of SAMP10 mice showed an increase in the expression of proteins, except for NR2B. Although there is some room for memory consideration regarding the involvement of transcription factors such as the cyclic AMP response element binding protein (CREB), which is essential for the process of memory consolidation, the results of behavioral experiments and memory-related protein expression in this study suggest the importance of GluR1 and phosphorylated CaMKII in maintaining learning and memory functions. In addition, when memory is consolidated, synaptic changes occur in excitatory neurons that use glutamate as a neurotransmitter. AMPA- and NMDA- type glutamate receptors, which play important roles in the memory processes that occur in the postsynaptic membrane, are thought to mediate some of these changes [60,70].

Although the detailed mechanism of how SAC affects learning memory is not clear, we speculate that SAC might produce useful changes on the action of AMPA- and NMDA-type receptors in the postsynaptic membranes and on the mechanism of memory formation. MGE was also found to contribute to equal or better maintenance of postsynaptic function. As with cultured hippocampal neurons, this phenomenon is considered to be a synergistic effect of the multiple other components contained in MGE. Our findings indicate that MGE and SAC are possibly involved in the regulation of synaptic plasticity through mechanisms that promote hippocampal neuronal differentiation, regulate the synaptic microenvironment, and suppress a decrease in memory-related proteins.

## 5. Conclusions

We suggest that MGE and SAC positively contribute to learning, memory formation, and the maintenance of young brain function. The results of this study were obtained using SAMP10 mice, which showed morphological changes similar to those in humans with mild memory and cognitive impairments during aging. Thus, MGE and SAC could be applied in foods to improve the accuracy of memory and judgment.

## Figures and Tables

**Figure 1 nutrients-12-01834-f001:**
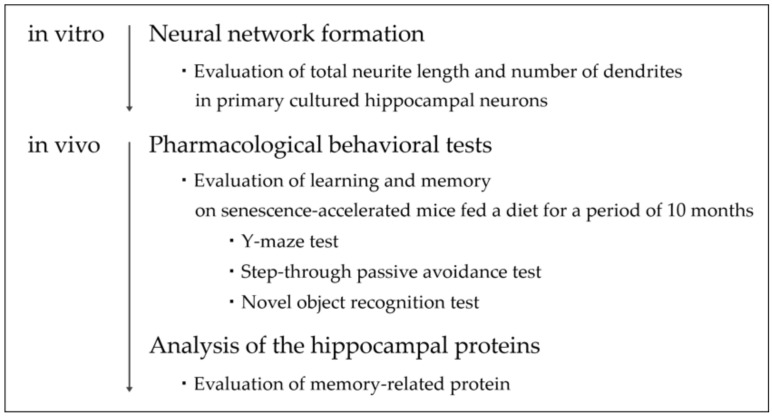
Flowchart of the study.

**Figure 2 nutrients-12-01834-f002:**
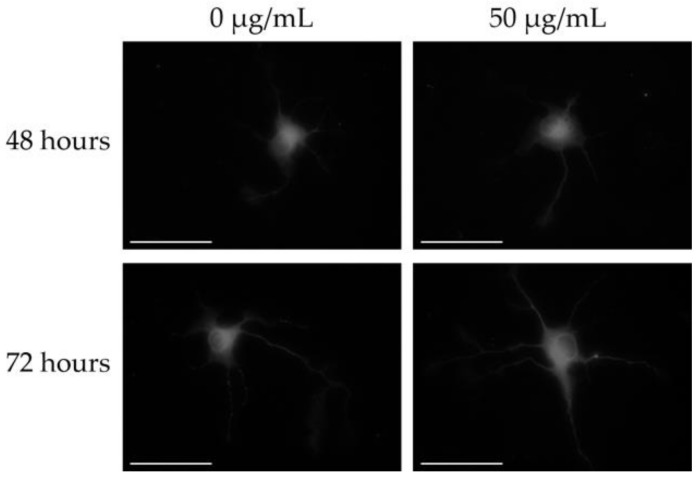
Images of MAP2-positive primary cultured hippocampal neurons treated with MGE. The neurons were treated with 0 (control) or 50 μg of MGE for 48 (day in vitro (DIV) 4) or 72 h (DIV 5). These images of MAP2-positive cells obtained by immunofluorescence staining were subjected to image conversion using the IN Cell Translator. Bars = 100 μm.

**Figure 3 nutrients-12-01834-f003:**
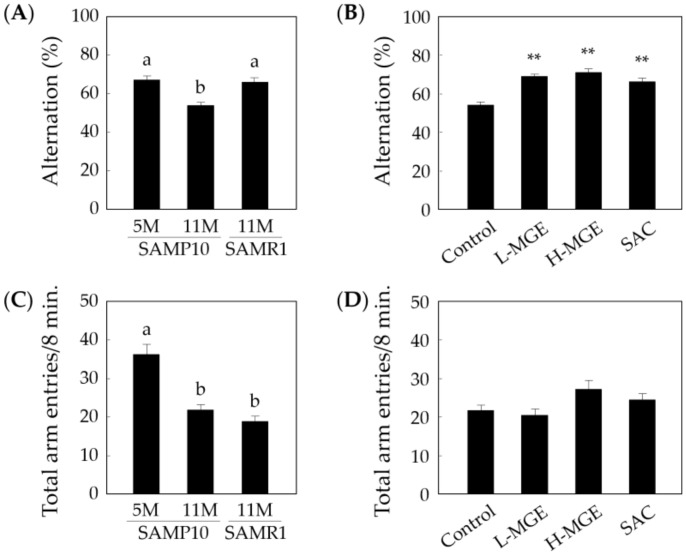
Effects of MGE and SAC on working memory in senescence-accelerated mice (SAM). The behavior of SAMP10 (5 and 11 months old) and SAMR1 (11 months old) was observed for 8 min in a Y-maze. The ratios of alternation (%, (**A**,**B**)) and total arm entries (**C**,**D**) were measured. (**B**,**D**) show the results for 11-month-old SAMP10. The L- and H-MGE groups consumed 0.20% and 1.0% of MGE in the diet (*w*/*w*), respectively. The SAC and H-MGE groups consumed the same amount of SAC. Each value represents the mean ± SEM (*n* = 8–17). ^a,b^; *p* < 0.05 (One-way ANOVA and Tukey–Kramer post-hoc test), **; *p* < 0.01 compared with the control group (One-way ANOVA and Dunnett’s post-hoc test). SAMP: senescence-prone inbred strains; SAMR: senescence-resistant inbred strains.

**Figure 4 nutrients-12-01834-f004:**
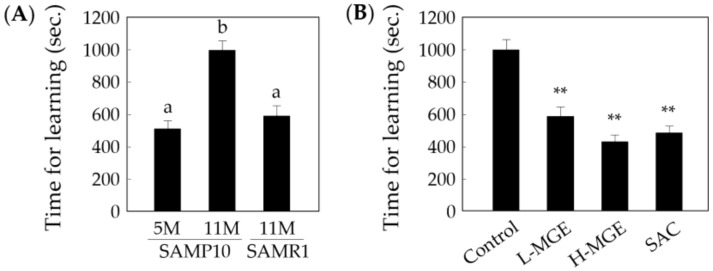
Effects of MGE and SAC on learning in SAM mice. (**A**) The learning time of SAMP10 (5 and 11 months old) and SAMR1 (11 months old) was examined using a step-through test system. The time needed for acquisition (**A**,**B**) of the avoidance response was measured. (**B**) Results for 11-month-old SAMP10 mice. The L- and H-MGE groups consumed 0.20% and 1.0% of MGE in the diet (*w*/*w*), respectively. The SAC and H-MGE groups consumed the same amount of SAC. Each value represents the mean ± SEM (*n* = 8–16). ^a,b^; *p* < 0.05 (One-way ANOVA and Tukey–Kramer post-hoc test), **; *p* < 0.01 compared with the control group (One-way ANOVA and Dunnett’s post-hoc test).

**Figure 5 nutrients-12-01834-f005:**
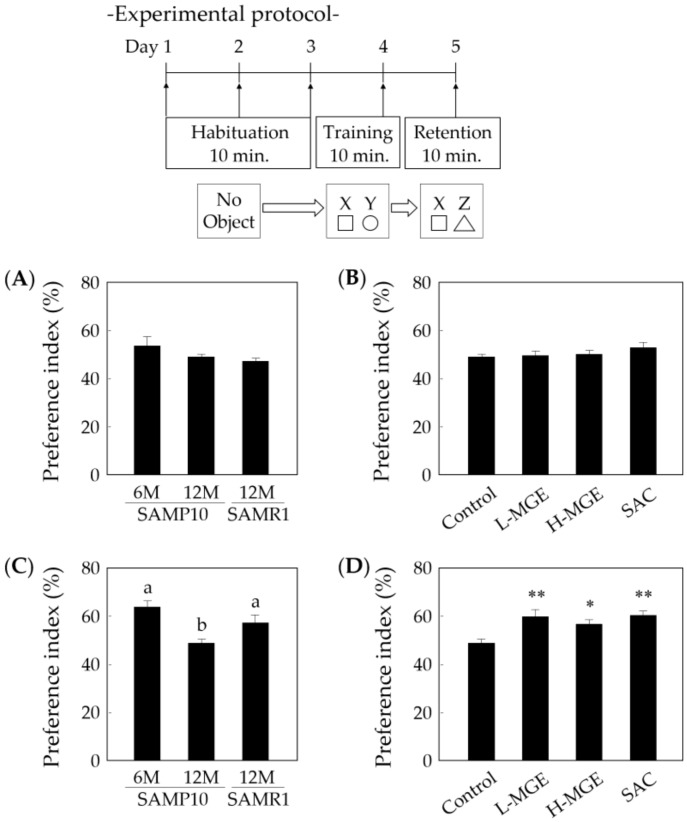
Effects of MGE and SAC on non-spatial memory in SAM mice. The exploratory preference (%) of SAMP10 (6 and 12 months old) and SAMR1 (12 months old) was observed in the novel object recognition test. The time spent exploring two objects was recorded for 10 min in the training sessions (**A**,**B**) and retention sessions (**C**,**D**). (**B**,**D**) show the results for 12-month-old SAMP10 mice. The L- and H-MGE groups consumed 0.20% and 1.0% of MGE in the diet (*w*/*w*), respectively. The SAC and H-MGE groups consumed the same amount of SAC. Each value is presented as the mean ± SEM (*n* = 8–15). ^a,b^; *p* < 0.05 (One-way ANOVA and Tukey–Kramer post-hoc test), *; *p* < 0.05, **; *p* < 0.01 compared with the control (One-way ANOVA and Dunnett’s post-hoc test).

**Figure 6 nutrients-12-01834-f006:**
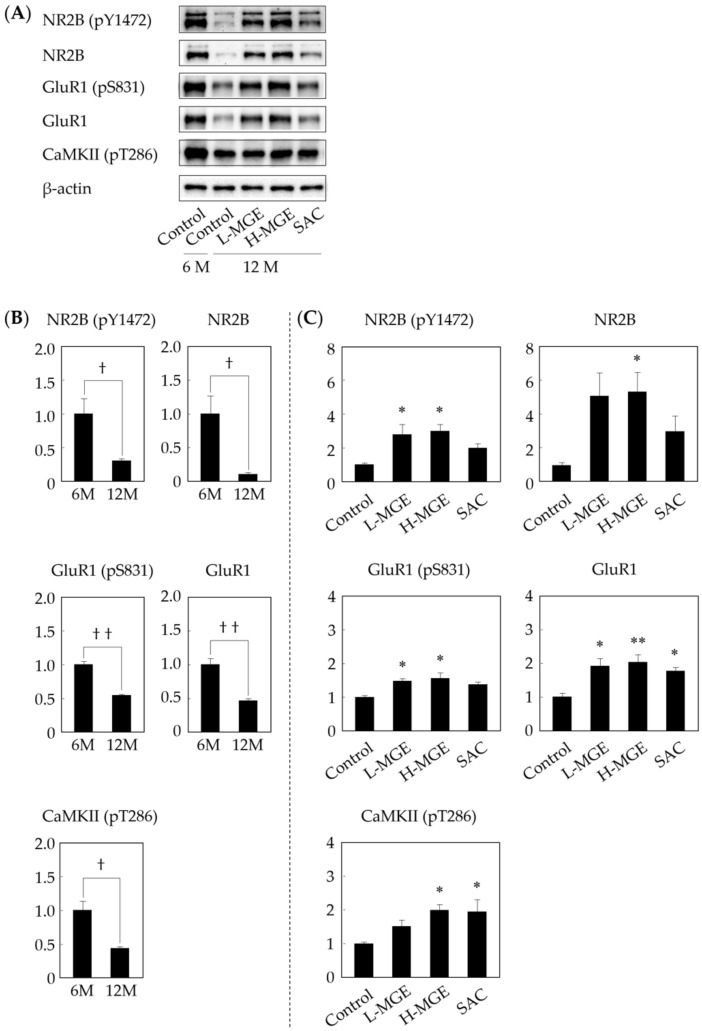
Effects of MGE and SAC on hippocampal proteins in SAMP10 mice. The molecular mechanism of memory was investigated by Western immunoblotting of hippocampal tissue after completion of the behavioral experiments (**A**). Protein levels for 6- and 12-months-old SAMP10 mice (**B**) and 12-months-old SAMP10 mice (**C**) are shown. Each value is presented as the mean ± SEM (*n* = 4). ^†^; *p* < 0.05, ^† †^; *p* < 0.01 (Student’s *t*-test), *; *p* < 0.05, **; *p* < 0.01 compared with the control group (One-way ANOVA and Dunnett’s post-hoc test).

**Table 1 nutrients-12-01834-t001:** Effects of increasing the total neurite length with mature garlic extract (MGE) and S-allylcysteine (SAC) on the morphology of primary cultured hippocampal neurons.

	Concentration	48 h	72 h
		Total Neurite Length, μm	Total Neurite Length, μm
MGE (μg/mL)	0	278.9 ± 7.24	362.9 ± 8.60
	5	306.7 ± 9.04 *	419.3 ± 9.53 **
	50	323.5 ± 8.77 **	430.7 ± 9.40 **
	500	392.8 ± 8.18 **	435.2 ± 9.30 **
SAC (ng/mL)	0	259.1 ± 6.85	342.9 ± 8.15
	10	300.3 ± 8.04 **	445.8 ± 12.9 **
	100	294.8 ± 7.08 **	433.5 ± 11.3 **
	1000	303.8 ± 7.56 **	398.1 ± 7.56 **

0 h: 212.5 ± 6.08 μm. The data are presented as the mean ± SEM (*n* = 102–144 from 3–5 independent experiments). *; *p* < 0.05, **; *p* < 0.01 compared with the concentration at 0 h at each time point (one-way ANOVA and Dunnett’s post-hoc test).

**Table 2 nutrients-12-01834-t002:** Effects of increasing the number of dendrites with MGE and SAC on the morphology of primary cultured hippocampal neurons.

	Concentration	48 h	72 h
		Number of Dendrites	Number of Dendrites
MGE (μg/mL)	0	5.29 ± 0.137	5.29 ± 0.186
	5	5.86 ± 0.176 *	7.16 ± 0.217 **
	50	6.06 ± 0.166 **	6.88 ± 0.196 **
	500	6.81 ± 0.207 **	7.92 ± 0.226 **
SAC (ng/mL)	0	5.34 ± 0.190	5.32 ± 0.148
	10	6.19 ± 0.166 **	7.35 ± 0.242 **
	100	6.20 ± 0.163 **	6.76 ± 0.203 **
	1000	5.88 ± 0.133	6.46 ± 0.180 **

0 h: 3.47 ± 0.127. The data are presented as the mean ± SEM (*n* = 102–144 from 3–5 independent experiments). *; *p* < 0.05, **; *p* < 0.01 compared with the concentration at 0 h at each time point (one-way ANOVA and Dunnett’s post-hoc test).

**Table 3 nutrients-12-01834-t003:** Effects of MGE and SAC on the memory of the avoidance response in the retention test for SAMP10 (6 and 12 months old) and SAMR1 (12 months old) mice.

Mice	Age	Number of Animals Success (Left) Failure (Right)		Memory Retention (%)
SAMP10	6 M (young)	8	0	100
SAMP10	12 M (old)	7	8	46.7
SAMR1	12 M (old)	7	4	63.6

**Table 4 nutrients-12-01834-t004:** Effects of MGE and SAC on the memory of the avoidance response in the retention test for 12-month-old SAMP10 mice.

Mice	Diet	Number of Animals Success (Left) Failure (Right)		Memory Retention (%)
SAMP10	Control	7	8	46.7
SAMP10	L-MGE	9	4	69.2
SAMP10	H-MGE	10	3	76.9
SAMP10	SAC	11	4	73.3

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
