# Peer review of "Improvement of Learning and Memory in Senescence-Accelerated Mice by S-Allylcysteine in Mature Garlic Extract"

_nutrients, 2020, doi:10.3390/nu12061834_

Round 1
Reviewer 1 Report
- The English use at times is confusing, particularly in the ‘Introduction” and “Results” section. For example, Line 333-334”: Old-SAMP10 mice fed H-MGE showed significant suppression of the decrease in the expression levels of all proteins. It is better to rewrite it as “Treatment with H-MGE significantly reversed the protein loss observed in Old-SAMP10 mice”.
- The objective of including SAC in the test is not very clear. The authors stated that SAC was given separately in an amount similar to that found in H-MGE. I presume this was done to see whether SAC or SAC along other constituents are responsible to the observed effect. Time and again SAC showed an inconsistent effect in the observed parameters, particularly in the in vitro studies. To see the aforementioned effect, not only treatment is compared with controls, but SAC should also be compared with H-MGE. Hence, I advise the authors to reframe the discussion so as it could reflect such issues rather than speculating other things. Title should also be modified accordingly.
- Fig 2: total arm entry was not different between treatment and control in old SAMP10 mice. However, alternation increased with treatment. What is the biological meaning of this finding? Is there any relationship between the decrease in alternation in the controls and their overall performance in locomotion (total arm entry)? I think issues such as this should stand out in the Discussion.
- Passive avoidance test (Figure 3 and 2): The way the parameters measured (Method section) and the data are presented (Results section) is a bit confusing. In the Method section, 300 sec was taken as a maximum crossover latency and latencies measured thereafter were subtracted from the maximum latency. Lower values show higher learning. Do data presented in Figure 3 obtained after subtracting from 300 sec. Why not the authors straightly present latency to enter the dark compartment and time spent in the bright compartment. The retention test was carried out after a long period of time. What would that signify?
- Figure 4 (Page 9): in the novelty test, treatment did not change learning (Figure 4B) and yet memory was tested and treatment showed improvement in memory. Does that mean memory can be formed without learning? Doesn't that negate the relationship between the two phenomena?
- Line 354-355: We suggest that a nerve growth factor-like effect of MGE and SAC was strongly involved….. this is pure speculation as the extracts were not run using these factors as standards. In fact, the lack of using a positive control should be cited as limitation of the study.
- Did you subject the same animals to the different behavioral experiments? If yes, what would be the influence of one test on the other? Western blot was performed using the hippocampal tissue. Hippocampus is not involved in encoding all types of memory. For example, in the passive avoidance test, one can have two types of memory: contextual (associated with the electric shock) and emotional (fear of the dark box). The former is encoded by hippocampus and the latter by amygdala. Several structures and different transmitter systems are also involved in novelty. Given these, is it justified to make inference based on measurement made using only the glutamate system?
- Phosphorylation of proteins, including those measured in the current study, marks synaptic plasticity. Wouldn’t it better to analyze the ratio of phosphorylated and unphosphorylated forms rather than the individual forms, if we talk of activation. There are two bands for NR2B in the blot. I presume the lower one is NR2B. What is the upper one?
- CamKII level was measured but Ser831 can be phosphorylated by CamKII as well as PKC. What is the rationale for excluding the measurement of PKC?
- AMPA receptor are known to be regulated by covalent modification. Phosphorylation of Ser 831 and Ser 845 modulate single channel conductance and opening probability of the receptor, respectively. Moreover, studies show that specific phosphorylation site modulated is dependent on the history of synaptic plasticity. In naïve synapses, LTP induction increases phosphorylation on Ser 831, while in previously depressed synapses LTP induction increases phosphorylation on Ser 845. Given the animals included appeared to be of having depressed synapses, would Ser 831 be the best marker?
Author Response
Response to Reviewer 1 Comments
First of all, we would like to thank you for spending time reviewing our manuscript. Your
constructive comments clearly helped us improve our manuscript significantly.
Point 1: The English use at times is confusing, particularly in the ‘Introduction” and “Results”
section. For example, Line 333-334”: Old-SAMP10 mice fed H-MGE showed significant
suppression of the decrease in the expression levels of all proteins. It is better to rewrite it as
“Treatment with H-MGE significantly reversed the protein loss observed in Old-SAMP10
mice”.
Response 1: I thank you for your advice. We have corrected the grammatical issues in the
manuscript by asking MDPI's English proofreading service, including your pointed sentence.
Point 2: The objective of including SAC in the test is not very clear. The authors stated that
SAC was given separately in an amount similar to that found in H-MGE. I presume this was
done to see whether SAC or SAC along other constituents are responsible to the observed
effect. Time and again SAC showed an inconsistent effect in the observed parameters,
particularly in the in vitro studies. To see the aforementioned effect, not only treatment is
compared with controls, but SAC should also be compared with H-MGE. Hence, I advise
the authors to reframe the discussion so as it could reflect such issues rather than speculating
other things. Title should also be modified accordingly.
Response 2: Thank you for your valuable comment. There are many reports that SAC
positively contributed to the various functions of central nervous system (CNS). Although
the detailed mechanism of how SAC affects learning memory has not been clear yet.
Therefore, we aim to evaluate the effect of SAC on learning and memory, not to determine
the contribution ratio of SAC to the effect of MGE. However, it is sure that the other
components in MGE has the similar positive effect.
According to your suggestion, the title has been modified as follows:
Title:
Improvement of Learning and Memory in Senescence-Accelerated Mice by
S-Allylcysteine in Mature Garlic Extract
Discussion part has been revised as follows:
(Page 13, lines 432-437)
Although the detailed mechanism of how SAC affects learning memory is not clear,
we speculate that SAC might produce useful changes on the action of AMPA- and
NMDA-type receptors in the postsynaptic membranes and on the mechanism of
memory formation. MGE was also found to contribute to equal or better
maintenance of postsynaptic function. As with cultured hippocampal neurons, this
phenomenon is considered to be a synergistic effect of the multiple other
components contained in MGE.
Point 3: Fig 2: total arm entry was not different between treatment and control in old
SAMP10 mice. However, alternation increased with treatment. What is the biological
meaning of this finding? Is there any relationship between the decrease in alternation in the
controls and their overall performance in locomotion (total arm entry)? I think issues such as
this should stand out in the Discussion.
Response 3: Thank you for your suggestion. Figure 2 has been changed to Figure 3.
The reason for comparing the number of total arm entries is to evaluate that there is no
difference in the amount of exercise and exploratory behavior, and that the number of entries
does not change so much for comparison. There is no difference in the locomotor activity
obtained from the number of arm entries in Fig. 3. It means that there was no tension or
relaxation in any group. The increased alternation rates shown by treating MGE or SAC
indicate that the memory of the arm before 2 times and the memory before 1 time is retained.
I modified the following sentences in Discussion and related citation in Reference.
(Page 12, lines 374-378 in Discussion)
The Y-maze is considered to be a hippocampus-dependent memory test, since it
evaluates spatial working memory, an index of short-term memory, through
assessing the continuous selection of arms [46,47]. In addition, it has the advantage
of providing a measure of locomotor activity of mice by counting the number of
arm entries [48].
(Page 16, lines 571-572 in Reference)
48. Hughes, R.N. The value of spontaneous alternation behavior (SAB) as a test of
retention in pharmacological investigations of memory. Neuroscience & Biobehavioral
Reviews. 2004, 28, 497-505.
(Page 12, lines 389-393)
In all pharmacological behavioral tests performed, SAMP10 mice fed MGE or SAC
showed significantly reduced learning and memory dysfunction and significant
improvements in learning and short- and long-term memory formation. In addition,
since the locomotor activity obtained from the number of arm entries did not differ
significantly in any group, there were no differences in the amount of exercise,
exploratory behavior, or motivation.
Point 4: Passive avoidance test (Figure 3 and 2): The way the parameters measured (Method
section) and the data are presented (Results section) is a bit confusing. In the Method section,
300 sec was taken as a maximum crossover latency and latencies measured thereafter were
subtracted from the maximum latency. Lower values show higher learning. Do data
presented in Figure 3 obtained after subtracting from 300 sec. Why not the authors straightly
present latency to enter the dark compartment and time spent in the bright compartment.
The retention test was carried out after a long period of time. What would that signify?
Response 4: I’m sorry for not explaining sufficiently. Figure 3 has been changed to Figure 4.
The reason for ‘Why not the authors straightly present latency to enter the dark compartment
and time spent in the bright compartment.’ is as follows:
In this measurement, the time from the light room to the dark room is measured up to 5
times in succession, and these are added together for evaluation. For example, as in the
example below, the time spent in a light room cannot be evaluated correctly.
A mouse: 1st 10 s, 2nd 300 s (end of test)
B mouse: 1st 10 s, 2nd 10 s, 3rd 10 s, 4th 10 s, 5th 10 s (end of test)
C mouse: 1st 10 s, 2nd 150 s, 3rd 300 s (end of test)
When these mice were evaluated in the time they were in the light room, B (50s) <A (310s)
<C (460s), which means that the time required for B mice to learn was shortest.
On the other hand, when evaluated by the value subtracted from 300 seconds, B (1450s)>
C (440s)> A (290s), and it can be evaluated that the time required for A mouse to learn is
the shortest and the learning ability is the highest.
A: (300-10) + (300-300) = 290
B: (300-10) + (300-10) + (300-10) + (300-10) + (300-10) = 1450
C: (300-10) + (300-150) + (300-300) = 440
The reason why the retention test was carried out after a long period of time is that we
examined if the mouse retained the memory "not to enter the darkroom" even after a
month.
I added the following sentence in Materials and Methods (2. 4. 2.).
(Page 4, lines 168-170)
The total time spent in the light chamber during each trial was deducted from 300
seconds and was considered the time needed for learning. The time taken for each
trial was totaled—the shorter the learning time, the higher the learning ability.
Point 5: Figure 4 (Page 9): in the novelty test, treatment did not change learning (Figure 4B)
and yet memory was tested and treatment showed improvement in memory. Does that mean
memory can be formed without learning? Doesn't that negate the relationship between the
two phenomena?
Response 5: Thank you for your comment. Figure 4 has been changed to Figure 5. In a
training session (Figure 5B), it is important to learn and memorize. Next, in retention sessions
(Figure 5D), if the memory learned in the training session is retained, the preference for novel
objects may be increased.
Point 6: Line 354-355: We suggest that a nerve growth factor-like effect of MGE and SAC was
strongly involved….. this is pure speculation as the extracts were not run using these factors
as standards. In fact, the lack of using a positive control should be cited as limitation of the
study.
Response 6: We agreed with your comment and modified these sentences in Discussion as
follows:
(Page 12, lines 363-368)
In this study, treatment of primary cultured hippocampal neurons with MGE or
SAC significantly increased the total neurite length and number of dendrites. SAC
showed a certain neurotrophic effect and further activated neurons through a
synergistic effect with multiple other components contained in MGE. We suggest
that the neurotrophic effects of MGE and SAC are strongly involved in the
enhancement of transmission efficiency and information-processing ability by
neural network formation.
Point 7: Did you subject the same animals to the different behavioral experiments? If yes,
what would be the influence of one test on the other? Western blot was performed using the
hippocampal tissue. Hippocampus is not involved in encoding all types of memory. For
example, in the passive avoidance test, one can have two types of memory: contextual
(associated with the electric shock) and emotional (fear of the dark box). The former is
encoded by hippocampus and the latter by amygdala. Several structures and different
transmitter systems are also involved in novelty. Given these, is it justified to make inference
based on measurement made using only the glutamate system?
Response 7: Yes. Same animals were performed all cases of pharmacological behavioral tests.
According to Paylor et al., a 1-2 day test interval showed no significant difference in
behavioral test results compared to a weekly interval (Physiol Behav. 2006, 87, 95-102). First,
Y-maze test with less stress on mice was performed. After a 24-hour interval, a passive
avoidance test with electric shock was conducted. After that, novel object recognition test
was conducted for a period of 10 days. Therefore, it is considered that the influence on each
test may be small.
In the passive avoidance test, although amygdala is important for fear conditioning, it is
reported that fear memory is dependent on the hippocampus, and AMPA receptor
translocation to hippocampal synapses is necessary for fear memory formation (PNAS. 2011,
108, 12503-12508). Therefore, we chose the hippocampal glutamate receptor.
Point 8: Phosphorylation of proteins, including those measured in the current study, marks
synaptic plasticity. Wouldn’t it better to analyze the ratio of phosphorylated and
unphosphorylated forms rather than the individual forms, if we talk of activation There are
two bands for NR2B in the blot. I presume the lower one is NR2B. What is the upper one?
Response 8: We wish to thank the reviewer for this comment. The ratio of band intensity of
NR2B (pY1472) to NR2B and GluR1 (pS831) to GluR1 were calculated. However, these
results are difficult to explain, and since long-term potentiation is important for the number
(amount) of receptors, especially in AMPA, we did not include these results.
The lower band corresponded to the inactivated NR2B (166 kDa), and the upper one
corresponded to the NR2B (pY1472) (180 kDa). Two bands were detected in a paper using
the same antibody (Nature neuroscience. 2012, 15, 1227-1235), which is similar to our results.
Point 9: CamKII level was measured but Ser831 can be phosphorylated by CamKII as well
as PKC. What is the rationale for excluding the measurement of PKC?
Response 9: Thank you for your comment. Since the amount of CaMKII autophosphorylation
could be evaluated by the phosphorylation capacity of AMPA receptor, we considered that
it is more suitable for the evaluation of postsynaptic function than PKC.
Point 10: AMPA receptor are known to be regulated by covalent modification.
Phosphorylation of Ser 831 and Ser 845 modulate single channel conductance and opening
probability of the receptor, respectively. Moreover, studies show that specific
phosphorylation site modulated is dependent on the history of synaptic plasticity. In naïve
synapses, LTP induction increases phosphorylation on Ser 831, while in previously
depressed synapses LTP induction increases phosphorylation on Ser 845. Given the animals
included appeared to be of having depressed synapses, would Ser 831 be the best marker?
Response 10: Since the S845 site of NR2B is phosphorylated in the steady state, the evaluation
of the S831 site phosphorylated by LTP was considered to be the best marker for evaluating
memory enhancement. We had no consideration of depression, which cause
dephosphorylation of serine 845 during long-term depression (LTD).

Reviewer 2 Report
Paper is good and worth of publishing , few errors

Reviewer 3 Report
The manuscript is in line with the currently popular trend in neuro-psychiatric medicine to look for the intervention that may alleviate some bothering symptoms of ageing process. Although presented data seems to support the conclusions and experiments have been conducted properly, there are few issues that I strongly encourage authors to address before the paper is ready to be published:
- Describe in details the study in rodents following in here and in other sections STROBE statements
- I sugest to add a flow chart to easier follow the protocol of the study
- Describe if/and how the gender of animals affect aging process and thus the results of present study
- How the allocation of animals looked like?
- How was the dose of extracts established? How did you monnitor the intake of the intervention
- Did you check the normality? Were all variable normally distributed?
Round 2
Reviewer 1 Report
The authors have addressed my concerns and can be accepted after attending minor typos in the MS.
